# Adsorption of Arsenic and Heavy Metals from Solutions by Unmodified Iron-Ore Sludge

**Khai M. Nguyen [1], Bien Q. Nguyen [1], Hai T. Nguyen [2] and Ha T.H. Nguyen [1,*]**

[1] VNU University of Science, Vietnam National University, 334 Nguyen Trai, Ha Noi 10053, Vietnam; nguyenmanhkhai@hus.edu.vn (K.M.N.); vio.quocbien@gmail.com (B.Q.N.)

[2] Faculty of Engineering and Information Technology, University of Technology, Sydney (UTS), P.O. Box 123, Broadway, Sydney, NSW 2007, Australia; hainguyennguyen92@gmail.com

[*] Correspondence: hoangha.nt@vnu.edu.vn or hoanghantvnu@gmail.com; Tel.: +84-243-558-7060

**Abstract:** Arsenic and heavy-metal-contaminated environments are a major concern due to their negative impacts on exposed people and ecosystems. In this study, sludge from an iron-ore processing area was used as an adsorbent to remove As, Mn, Zn, Cd, and Pb from aqueous solutions. The adsorption capacity of target adsorbates was investigated in batch experiments of both single- and mixed-metal solutions. The batch studies show that the maximum Langmuir adsorption capacities of the heavy metals onto the adsorbent occurred in the order Pb > As > Cd > Zn > Mn, and ranged from 0.710 mg/g to 1.113 mg/g in the single-metal solutions and from 0.370 mg/g to 1.059 mg/g in the mixed-metal solutions. The results of the kinetic experiments are consistent with pseudo-first-order and pseudo-second-order models, with a slightly better fit to the latter. Adsorption performances indicate that iron-ore sludge can simultaneously adsorb multiple metal ions and is a promising adsorbent for the removal of toxic pollutants from water.

**Keywords:** adsorption; arsenic; heavy metal; removal; unmodified iron ore sludge

## 1. Introduction

The pollution of wastewater with arsenic (As) and heavy metals is one of the most serious environmental problems worldwide. Considerable amounts of wastewater, generated from anthropogenic activities such as mining and smelting, fertilizer production, agriculture, and battery manufacturing, pose a high risk to the environment, ecosystems, and human health [1,2]. A variety of treatment methods have been applied to eliminate heavy metals from water, including coagulation [3], adsorption [4], ion exchange [5], electrocoagulation [6], and biological processes [7,8]. Of these, adsorption is considered the most cost-effective [9] when using sorbents that require little processing, are abundant in nature, or are by-products or waste materials from industry [10–12].

Solid wastes from mining activities have been assessed as low-cost adsorbents for wastewater purification [12–17]. A variety of solid wastes have been used for water treatment, including clay-bearing mining waste [18], red mud [13,19–21], coal mine-drainage sludge [22], iron-ore slime [17], and waste mud from copper mines [16]. The use of mining waste is considered to be an environmentally-friendly technique because both solid waste and contaminated water are treated and could result in waste-free production [12,14,16,23,24].

Vietnam's mining sector is the third largest contributor to the national gross domestic product (GDP) [25]. An increasing demand for iron in the steel and construction industries, and the abundance of iron deposits in Vietnam [26] have generated a large volume of solid waste. Failure of the sludge reservoir at the Ban Cuon iron-ore processing area, Bac Kan Province on July 20, 2014 damaged agricultural land and drainage infrastructure of the Ngoc Phai commune [27]. This event highlights an

urgent need for the management and treatment of such solid wastes. The use of iron-ore sludge for water treatment is potentially one solution; however, there are few studies on the capacity of iron-ore sludge to adsorb As and heavy metals or their potential use for water purification.

The aim of this research was to determine the adsorption behavior of As, Mn, Zn, Cd, and Pb in batch experiments using iron-ore drainage sludge collected from an iron processing area in northern Vietnam.

## 2. Materials and Methods

### 2.1. Preparation of Adsorbent and Solutions

The adsorbent applied in this study was collected from the Ban Cuon iron processing area, Cho Don District, Bac Kan Province, northern Vietnam (Ban Cuon iron-ore sludge, SBC). It is an iron-manganese ore formed by oxidation of the original ore-bodies (containing magnetite, siderite, pyrite, and pyrotine) or ultramafic rocks [28]. The ore mineral assemblage includes magnetite (63%), quartz (30.6%), and chlorite (6.3%) [29]. The iron content of ore ranges from 38.0% to 67.6% with an average of 59.3% [29].

Ten kilograms of iron-ore sludge were selected to be used as an adsorbent and dried using a NIIVE OVER KD200 oven at 80–105°C until constant weight was achieved. The dried materials were ground using an MRC Laboratory Equipment Manufac User and sieved to obtain particles ranging in size from 0.16 to 0.25 mm [30].

Four divalent cationic metals ($Mn^{2+}$, $Pb^{2+}$, $Zn^{2+}$, and $Cd^{2+}$) and As (V), were selected as target adsorbates to determine the adsorption characteristics of the adsorbent. These metals were prepared individually by diluting $Mn(NO_3)_2$, $Cd(NO_3)_2 \cdot 4H_2O$, $Pb(NO_3)_2$, $Zn(NO_3)_2$, and $Na_2HAsO_4 \cdot 7H_2O$ (Kanto Chemical Co. Inc., Japan) with Milli-Q water to obtain the desired concentrations for the batch experiments.

### 2.2. Experiments

#### 2.2.1. Point of Zero Charge ($pH_{PZC}$)

The adsorbent was added to 100 mL solutions with an initial pH ranging from 3.0 to 7.0. The pH was adjusted using 0.1 M NaOH and 0.1 M $HNO_3$. The suspensions were agitated in a flat shaker at a shaking speed of 120 rpm at room temperature (25 $\pm$ 2 °C) for 24 h. At the end of the period, the pH (equilibrium pH) was determined. The $pH_{PZC}$ was calculated using the following equation:

$$\Delta pH = pH_{(1)} - pH_{(2)} \tag{1}$$

where $pH_{(1)}$ is the pH value before the experiment and $pH_{(2)}$ presents the pH value after the experiment. The intersection of the connecting points line with the horizontal axis at the point $\Delta pH = 0$ indicates $pH_{PZC}$.

#### 2.2.2. Adsorption Study

The adsorption study of the tested contaminants was conducted in batch experiments of both single- (As, Mn, Zn, Cd, or Pb) and mixed-metal (Mn, Zn, Cd, and Pb) solutions. In the single-metal experiments, nitrate salt solutions of Mn, Zn, Cd, and Pb and $Na_2HAsO_4.7H_2O$ (Kanto Chemical Co. Inc., Japan) were separately diluted by Milli-Q water to obtain the desired concentrations of As and heavy metals. In the mixed-metal experiments, these nitrate salt solutions of Mn, Zn, Cd, and Pb were simultaneously prepared. The mixed-metal solutions excluded As due to the possible reaction between cations and anions in the same solution.

Different amounts of the adsorbent (10, 20, 40, and 80 g/L) were used to assess the effect of doses on adsorption. In the adsorption kinetic and equilibrium adsorption experiments, 2 g of adsorbent was added to 100 mL of the aqueous adsorbate solution in a 125 mL plastic flask. The initial pH values of

the solutions were adjusted by adding 0.1 M HCl and 0.1 M NaOH solutions to reach the desired value. The flask was covered with Parafilm and shaken at 120 rpm on Daihan Labtech LSI-2 orbital shaker at room temperature ($25 \pm 2°C$) for the desired time interval. After the predetermined time period had elapsed, the mixture of adsorbent and adsorbate was immediately separated using a 0.45 μm syringe filter. A 15 mL sample of the filtered solution was collected and diluted (if necessary) for elemental analysis.

The amount of adsorbate that was adsorbed at equilibrium, $q_e$ (mg/g), or at time t, $q_t$ (mg/g), was calculated by the following mass balance Equations (2) and (3):

$$q_e = \frac{(C_o - C_e)V}{m} \tag{2}$$

$$q_t = \frac{(C_o - C_t)V}{m} \tag{3}$$

where $C_o$ (mg/L) $C_e$ (mg/L) and $C_t$ (mg/L) are the adsorbate concentrations at beginning equilibrium and time t, respectively; m (g) is the mass of used adsorbent; and V (L) presents the volume of the adsorbate solution.

The adsorption kinetic experiments were conducted on single- and mixed-metal solutions with heavy metal concentrations of 20 mg/L. Glass flasks containing 100 mL of the heavy metal solutions were agitated in a flat shaker at 120 rpm at room temperature ($25 \pm 2°C$) with a constant pH of 5.5. The kinetic studies were conducted for specific durations ranging from 10 to 1440 min.

Adsorption equilibrium experiments were conducted in various initial concentrations of heavy metals (0–50 mg/L). Approximately 2 g of adsorbent was added to 100 mL of solution with predetermined adsorbate concentration in a 125 mL Plastic flask. The flasks were immediately covered with a parafilm and shaken at 120 rpm at the same conditions of kinetic experiments.

Desorption efficiency analysis of the target metal ions was also performed using the adsorbent from the isotherm experiments that had an initial metal concentration of 50 mg/L. This adsorbent was added to 100 mL of solutions with different pH values (4.0, 5.5, 7.0, and 9.0) and was agitated in a flat shaker at 120 rpm at $25 \pm 2$ °C for 24 h.

All experiments were conducted in duplicate, and the resulting data were averaged. If the bias of the repeated experiment exceeded 15%, a third run was conducted.

### 2.3. Analytical Methods

Mineral compositions of the adsorbent were determined using a Siemens D5005 X-ray Diffraction (XRD) on powder samples. The XRD was equipped with a Cu (Kα1,2) target at 40 kV and 30 mA with a setting of 3–70° (2θ) and a step time 0.02° (2θ). Surface area analysis was undertaken using a Micromeritics Gemini VII 2390 surface area analyzer and the Brunauer–Emmett–Teller method (BET) was applied to estimate the specific surface area ($S_{BET}$). Cation exchange capacity (CEC) was determined using a Mütek PCD-05 particle charge detector (PCD). The functional groups available on the adsorbent's surface were identified using a Thermo Scientific Nicolet iS5 Fourier transform infrared spectrometer (FTIR). Morphological characteristics and surficial element composition of the adsorbent were obtained using a Thermo Scientific Quanta 650 scanning electronic microscope (SEM).

The suspensions in the solutions were separated using a 0.45 μm nylon syringe filters (Cole-Parmer). Elemental analysis was performed using an Agilent 240FS atomic absorption spectrometer (AAS) with hydride generation accessory VGA77.

All measurements were conducted at Key Laboratory of Geoenvironment and Climate Change Response (GEO-CRE), VNU University of Science, Vietnam National University, Hanoi, Vietnam.

## 3. Results and Discussion

### 3.1. Characteristics of Materials

The results of the XRD analysis of the SBC show high contents of quartz (39%), goethite (20%), kaolinite (11%), magnetite (11%), and muscovite (10%) and low contents of illite (7%), hematite (3%), and pyrite (1%). The SEM images show that the surface of SBC is porous and it has a large surface area with many fibers that facilitate ion adsorption processes on the surface of the material (Figure 1).

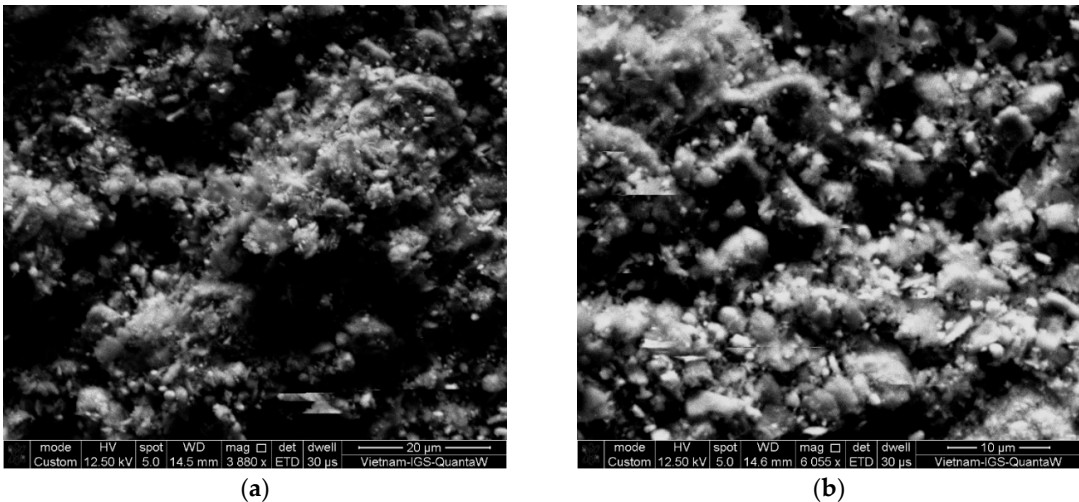

(**a**)  (**b**)

**Figure 1.** SEM images of iron-ore sludge at 3880× (**a**) and 6055× (**b**) magnifications.

The adsorbent was also characterized by FTIR spectroscopy, as shown in Figure 2. The obtained spectra has peaks at 3696 cm$^{-1}$ and 3619 cm$^{-1}$ that are probably from hydroxyl groups and O–H bending (Figure 2). The peaks at 1030 cm$^{-1}$ and 798 cm$^{-1}$ indicate the presence of Si–O–Si and Si–OH groups (Figure 2). The presence of large amounts of silica and hydroxyl groups could enhance the adsorption capacity of the adsorbent [31].

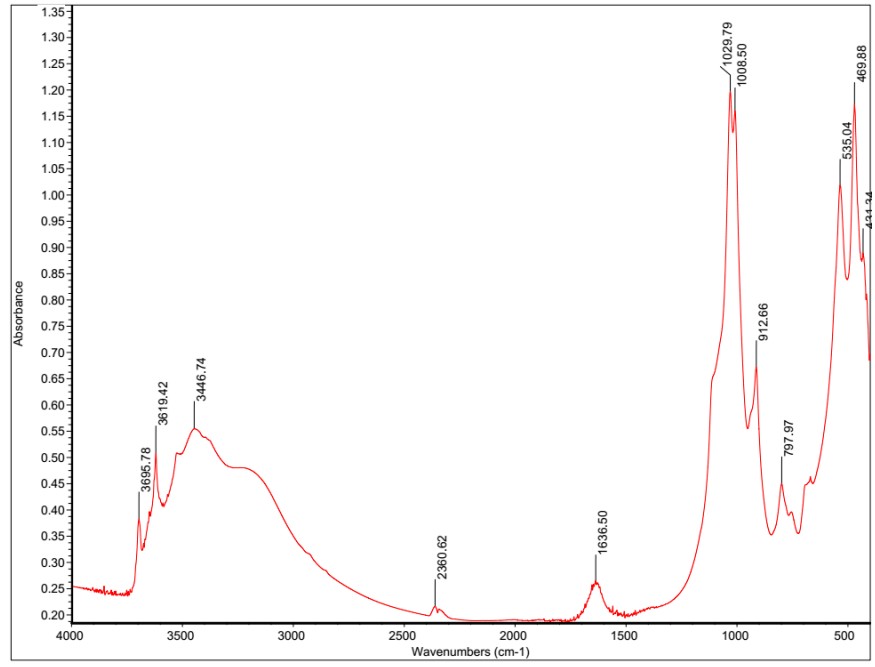

**Figure 2.** FTIR pattern of iron-ore sludge.

The specific surface area ($S_{BET}$) and cation exchange capacity (CEC) of an adsorbent can be used as indicators of adsorption capacity. An adsorbent with high $S_{BET}$ and CEC values is expected to have a high binding capacity for potentially toxic metals in aqueous solutions [32]. The $S_{BET}$ and CEC values of SBC were 47.4 m$^2$/g and 75 mmol$_{c(-)}$.Kg$^{-1}$, respectively. The $S_{BET}$ and CEC values of this adsorbent are higher than those reported for other materials such as laterite [33,34], clay [35,36], and red mud [19] (Table 1).

**Table 1.** Comparison specific surface area ($S_{BET}$) and cation exchange capacity (CEC) of iron-ore sludge with other materials.

|  | $S_{BET}$ (m$^2$/g) | CEC (mmol$_{c(-)}$.Kg$^{-1}$) | Authors |
|---|---|---|---|
| SBC | 47.4 | 75 | This study |
| Laterite | 15.365 | - | [33] |
| Laterite (Hanoi, Vietnam) | - | 66 | [34] |
| Clay | 10–20 | 15–75 | [35,36] |
| Red mud | 30 | 37 | [19] |

The point of zero charge (PZC) is defined as the solution conditions under which the surface charge density equals zero. The pH$_{PZC}$ of the material is 5.0 (Figure 3). Therefore, the adsorbent surface is positively charged at pH < 5.0 and becomes negatively charged at pH > 5.0. For pH values of <5.0, adsorption is hindered by repulsive electrostatic interactions between the metal ions and positively charged functional groups [37].

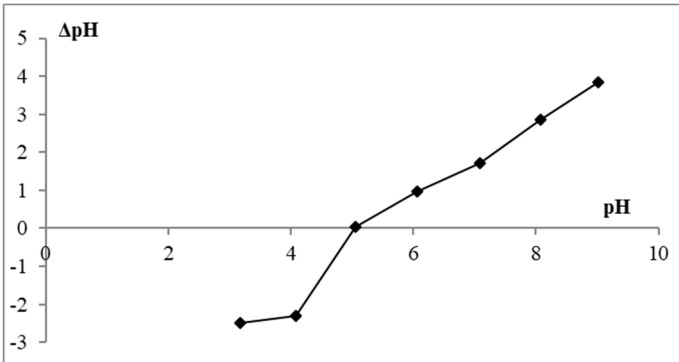

**Figure 3.** Zero charge point pH$_{PZC}$ of iron-ore sludge.

### 3.2. Effect of pH on Adsorption

The effects of pH on adsorption are shown in Figure 4. The adsorption capacity of SBC was strongly dependent on the pH of the solutions (pH$_{solution}$). The amount of target cations adsorbed by the adsorbent ($q_e$; mg/g) increased remarkably when the solution pH increased, with the exception of As. This is due to a decrease in competitive adsorption between H$^+$ ions and adsorbate cations for active sites on the surface of the adsorbent. The dependence of adsorption on pH is in agreement with the pH$_{PZC}$ and pH$_{solution}$ in the experiment (pH = 5.5). This result indicates that electrostatic attraction was the main mechanism controlling adsorption of the target elements.

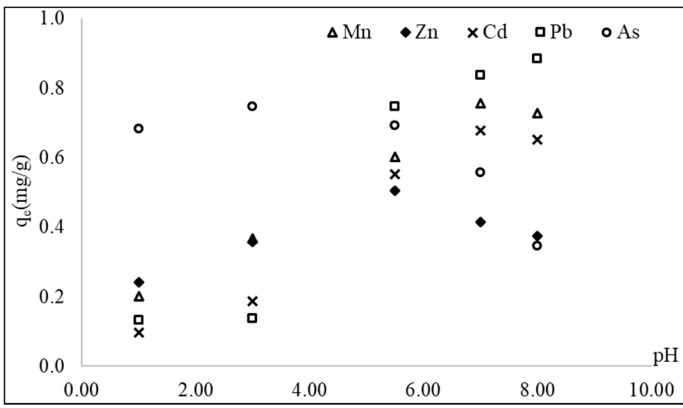

**Figure 4.** Effect of solution pH on the adsorption.

The results also demonstrate that the adsorption tendencies of Pb, Cd, Zn, and Mn were different to those of As (Figure 4). The adsorption of As and heavy metals can be explained by the following processes. Firstly, complexation occurs at pH $\geq$ 7 [38,39], and these complexes react with other cations in the solution, resulting in co-precipitation. Secondly, when the $pH_{solution}$ exceeds the $pH_{PZC}$ of the adsorbent, the surface is negatively charged, creating favorable conditions for cation adsorption [37]. Thirdly, as mentioned above, the increase in $pH_{solution}$ causes a decrease in competitive adsorption between $H^+$ ions and adsorbate cations.

### 3.3. Effect of Doses of the Adsorbent on Adsorption

The effect of different doses of the adsorbent on removal of As and heavy metals is shown in Figure 5. The results demonstrated higher removal efficiencies of As and Pb in the single-metal experiments and Pb in mixed-metal experiments than those of Zn, Cd, and Mn (Figure 5). An increase in doses of As and heavy metals causes an increase of metal removal from solutions. However, a slow increase in the metal removal efficiencies at doses of 40 and 80 g/L indicates that 20 g/L is an optimal dose for removal of As and heavy metals. Accordingly, dose of 20 g/L was selected in the batch adsorption kinetic and adsorption equilibrium experiments.

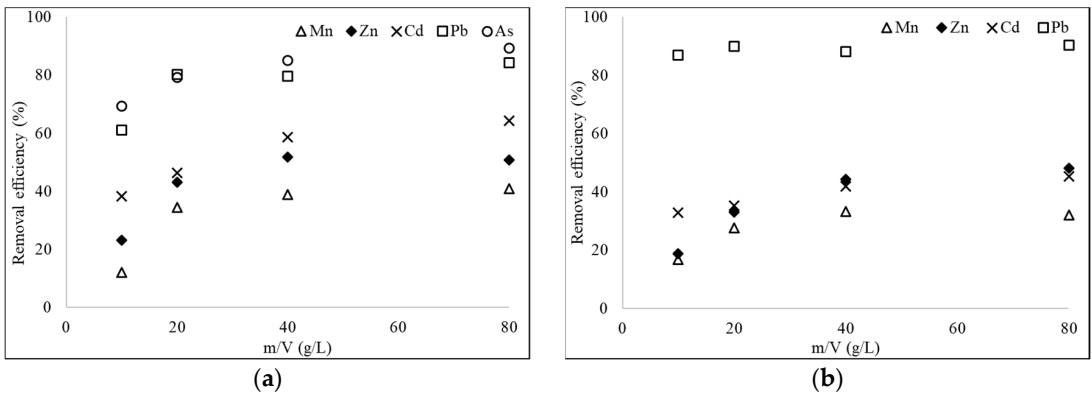

**Figure 5.** Effect of doses of the adsorbent on the adsorption As and heavy metals from (**a**) single-metal experiments and (**b**) mixed-metal experiments.

### 3.4. Batch Adsorption Kinetics

The adsorption kinetic experiments were conducted to assess the rate of metal adsorption onto the adsorbent. The adsorption capacity increased with contact time. Equilibrium was reached after 6 h in the single-metal experiments, and after 1h in the mixed-metal experiments (Figure 6).

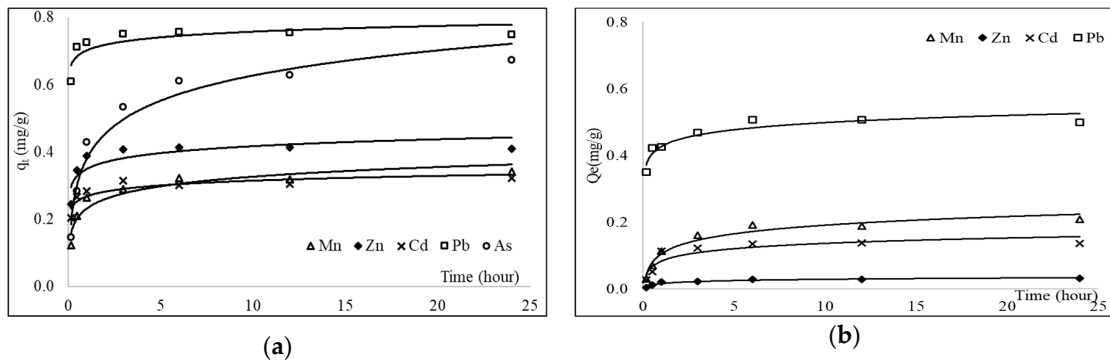

**Figure 6.** Effect of contact time on the adsorption of As and heavy metals from (**a**) single-metal experiments and (**b**) mixed-metal experiments.

In this study, two kinetic models were applied to mathematically describe the intrinsic adsorption constants from both the single- and mixed-metal experiments. The non-linearized forms of the pseudo-first-order model [40] and pseudo-second-order model [41] are expressed in the following equations:

$$q_t = q_e \left( 1 - e^{-k_1 t} \right) \tag{4}$$

$$q_t = \frac{q_e^2 k_2 t}{1 + k_2 q_e t} \tag{5}$$

where $k_1$ (min$^{-1}$) and $k_2$ (g·mg$^{-1}$·min$^{-1}$) are the rate constants of the pseudo-first- and pseudo-second-order models, respectively; $q_e$ (mg/g) and $q_t$ (mg/g) are the amounts of adsorbate uptake per mass of adsorbent at equilibrium and time t (min), respectively.

The adsorption kinetic data from the single- and mixed-metal experiments are consistent with the pseudo-first- and pseudo-second-order models (Table 2). The relative kinetic parameters demonstrate that the coefficient ($R^2$) values of the second-order kinetic model (0.881–1.000) are superior to those of the first-order kinetic model (0.596–0.969), with the exception of Cd. Therefore, the pseudo-second-order model is more suitable for explaining the kinetic behavior of heavy metals in the adsorbent of this study, indicating that the adsorption process might be controlled by chemisorptions processes [42,43].

**Table 2.** Relative kinetic parameters (calculated by the non-linear method) for the adsorption of Mn, Zn, Cd, Pb, and As onto the adsorbent from single- and mixed-metal experiments.

| Heavy Metals | Pseudo-First-Order Model | | | Pseudo-Second-Order Model | | |
|---|---|---|---|---|---|---|
| | $K_1$ (min$^{-1}$) | $q_e$ (mg/g) | $R^2$ | $K_2$ (g·mg$^{-1}$·min$^{-1}$) | $q_e$ (mg/g) | $R^2$ |
| Single-metal experiments | | | | | | |
| Mn | 0.035 | 0.318 | 0.931 | 0.121 | 0.345 | 0.957 |
| Zn | 0.066 | 0.406 | 0.897 | 0.653 | 0.427 | 0.941 |
| Cd | 0.125 | 0.305 | 0.995 | 0.565 | 0.321 | 0.940 |
| Pb | 0.177 | 0.737 | 0.902 | 0.451 | 0.761 | 0.947 |
| As | 0.018 | 0.641 | 0.954 | 0.039 | 0.692 | 0.985 |
| Mixed-metal experiments | | | | | | |
| Mn | 0.013 | 0.202 | 0.969 | 0.073 | 0.220 | 0.991 |
| Zn | 0.027 | 0.027 | 0.882 | 0.862 | 0.030 | 1.000 |
| Cd | 0.028 | 0.110 | 0.956 | 0.512 | 0.120 | 0.778 |
| Pb | 0.124 | 0.460 | 0.596 | 0.483 | 0.486 | 0.881 |

Based on the value of $k_2$, the adsorption rate of contaminant cations onto the adsorbent occurred in the following order: Zn > Cd > Pb > Mn > As for both single- and mixed-metal experiments. This result is in agreement with the order reported by Nguyen et al. [44] for zeolite (Zn > Cd > Pb).

### 3.5. Batch Equilibrium Adsorption

In this study, the adsorption isotherms for the single- and mixed-metal experiments show that the adsorption capacity decreased in the following order Pb > As > Mn > Cd > Zn (Figure 7). The Langmuir (Equation (6)) and Freundlich (Equation (7)) models were employed to describe the adsorptive behavior of selected adsorbates onto the adsorbent. To minimize the respective error functions, the non-linear optimization technique was applied in calculating the adsorption parameters from these models:

$$q_e = \frac{Q^0_{max} K_L C_e}{1 + K_L C_e} \tag{6}$$

$$q_e = K_F C_e^{\frac{1}{n}} \tag{7}$$

where $q_e$ and $C_e$ are obtained from Equation (2); $Q^o_{max}$ (mg/g) is the maximum saturated monolayer adsorption capacity of the adsorbent; $K_L$ (L/mg) is the Langmuir constant related to the affinity between the adsorbent and adsorbate; $K_F$ [(mg/g)/(mg/L)$^n$] is the Freundlich constant, describing the intensity of adsorption; $\frac{1}{n}$ (dimensionless; $0 < n < 10$) is a Freundlich intensity parameter, implying the magnitude of the adsorption driving strength or surface heterogeneity.

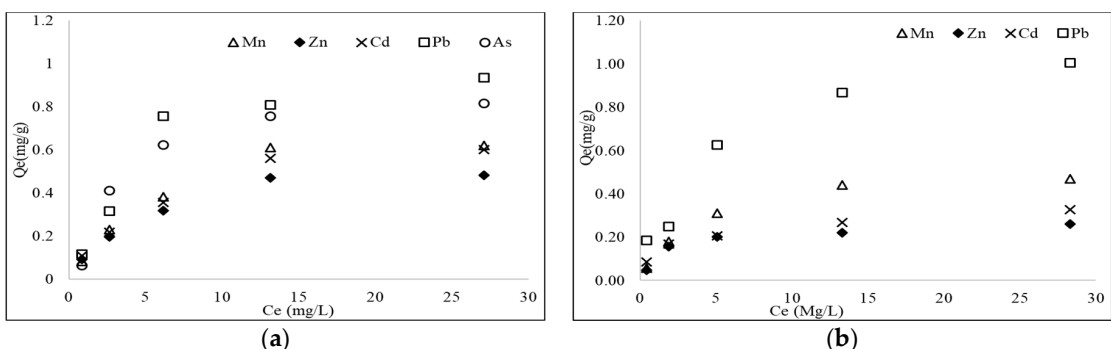

**Figure 7.** Adsorption isotherm of As and heavy metals onto the adsorbent from (**a**) single-metal experiments and (**b**) mixed-metal experiments.

The parameters of the adsorption models are listed in Table 3. The $R^2$ values of the Langmuir model (0.902–0.993) are higher than those of the Freundlich model (0.833–0.949), indicating that the adsorption characteristics of the contaminants in the adsorbent are adequately described by the Langmuir model. The data are consistent with the Langmuir adsorption model, suggesting that the adsorption sites were homogeneous with monolayer adsorption coverage [42].

**Table 3.** Adsorption isotherm parameters for As and heavy metals.

| Heavy Metals | Langmuir Model | | | Freundlich Model | | |
|---|---|---|---|---|---|---|
| | $Q_{max}^o$ (mg/g) | $K_L$ (L/mg) | $R^2$ | $K_F$ (mg/g)/(mg/L)n | *n* | $R^2$ |
| **Single-metal experiments** | | | | | | |
| Mn | 0.710 | 0.154 | 0.951 | 0.143 | 0.459 | 0.874 |
| Zn | 0.745 | 0.121 | 0.952 | 0.124 | 0.477 | 0.894 |
| Cd | 0.771 | 0.154 | 0.980 | 0.157 | 0.431 | 0.936 |
| Pb | 1.305 | 0.131 | 0.923 | 0.222 | 0.476 | 0.833 |
| As | 1.113 | 0.455 | 0.993 | 0.220 | 0.426 | 0.843 |
| **Mixed-metal experiments** | | | | | | |
| Mn | 0.370 | 0.217 | 0.975 | 0.100 | 0.350 | 0.854 |
| Zn | 0.447 | 0.263 | 0.955 | 0.109 | 0.402 | 0.949 |
| Cd | 0.484 | 0.338 | 0.982 | 0.136 | 0.229 | 0.910 |
| Pb | 1.059 | 1.913 | 0.902 | 0.577 | 0.261 | 0.870 |

The order of Langmuir maximum adsorption capacity for the single-metal experiments was Pb > As > Cd > Zn > Mn, with values of 1.305, 1.113, 0.771, 0.745, and 0.710 mg/g, respectively. Under the same experimental conditions, the $Q_{max}^o$ values in the mixed-metal experiments exhibited the following order: Pb > Cd > Zn > Mn, with values of 1.059, 0.484, 0.447, and 0.370 mg/g, respectively. The order of adsorption capacities might reflect the ionic radius of heavy metals ($Pb^{2+}$, $Cd^{2+}$, $Mn^{2+}$, $Zn^{2+}$), which affects the charge density of ions. The greater the cation radius, the smaller the charge density, and vice versa. The cation radius of the metals in the study are as follows: rPb(1.2 Å) > rCd (0.97Å) > rZn (0.74Å) > rMn (0.67Å) [45]. In addition, it is noted that the adsorption capacity of the heavy metals in the single-metal experiments is higher than that in the mixed-metal experiments, implying competition among heavy metals in the same solution [8].

The adsorption ability of heavy metals by SBC in this study can be explained by the specific surface area (BET), cation exchange capacity (CEC), presence of clay minerals (e.g., kaolinite and illite) [36], presence of large amounts of silica (Si–OH) [46,47] and O–H bending [46,48]. In addition, high contents of goethite (20%) may result in adsorption of As by the adsorbent [49,50].

The maximum Langmuir adsorption capacity ($Q_{max}^o$) of SBC is compared with that of other adsorbents in Table 4. The adsorption capacity of SBC was higher than that of laterite [34,51] and red mud [52] for As; laterite (OBY) for Pb [51]; and laterite (Tam Duong) for Zn, Cd, and Mn [34] (Table 4). However, the maximum adsorption capacity of SBC was lower than that of some clays and modified materials such as δ-FeOOH [53], kaolinite [36], montmorillonite [54,55], glucose AC [56], activated carbon [57], and ICZ (iron-coated zeolite) [44] (Table 4). Of note, the modification of raw materials increases their adsorption capacity [10,15,58]. Therefore, suitable modification of SBC should be performed to increase its adsorption capacity.

**Table 4.** Comparison of maximum adsorption capacity ($Q^o_{max}$) of some adsorbents

| Heavy Metals | Adsorbent | $Q^o_{max}$ (mg/g) | Authors |
|---|---|---|---|
| As | SBC | 1.113 | This study |
| | Laterite (Tam Duong) | 0.756 | [34] |
| | Laterite (OBY) | 0.702 | [51] |
| | Laterite soil | 1.384 | [33] |
| | Modified red mud | 1.08 | [52] |
| | δ-FeOOH | 37.3 | [53] |
| Pb | SBC | 1.305 | This study |
| | Laterite (Tam Duong) | 1.553 | [34] |
| | Laterite (OBY) | 0.658 | [51] |
| | Activated carbon | 20.7 | [57] |
| | Kaolinite | 4.730 | [36] |
| | Montmorillonite | 31.1 | [55] |
| | Glucose AC | 28.2 | [56] |
| Cd | SBC | 0.771 | This study |
| | Laterite (Tam Duong) | 0.397 | [34] |
| | Montmorillonite | 4.700 | [54] |
| | | 30.7 | [55] |
| | Activated carbon | 17.8 | [57] |
| | Modified biosorbents | > 45.4 | [32] |
| Zn | SBC | 0.745 | This study |
| | Laterite (Tam Duong) | 0.281 | [34] |
| | Activated carbon | 19.9 | [57] |
| | ICZ (Iron-coated zeolite) | 6.22 | [44] |
| Mn | SBC | 0.710 | This study |
| | Laterite (Tam Duong) | 0.143 | [34] |

## 3.6. Desorption of As and Heavy Metals at Different pH Values

In both the single- and mixed-metal desorption experiments the desorption rates of the metals were in the following order: Pb < As < Zn < Cd < Mn (Figure 8). The removal rate of Pb was 0.9%–3.0%, whereas that of Mn was 16.4%–27.8%. The desorption efficiency of metal ions decreased with increasing pH$_{solution}$, with the exception of As. Ion exchange and complexation could be involved in metal desorption from the adsorbent. This result is in agreement with previous studies [32,59].

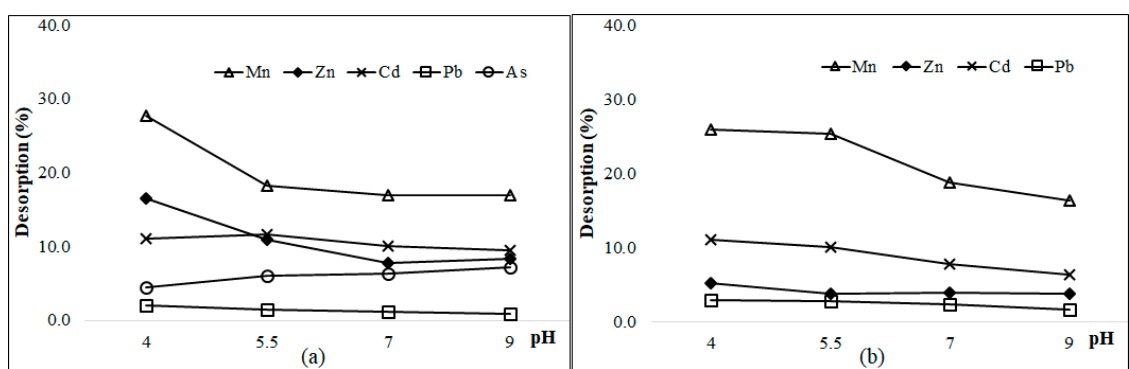

**Figure 8.** Desorption efficiency (%) of metal ions at various pH values from (**a**) single-metal experiments and (**b**) mixed-metal experiments.

The desorption efficiencies of the metal ions in this study are lower than those reported previously [32,44], possibly due to the low pH in the previous studies. The low desorption rates of SBC at pH values of 4.0–9.0 indicate a low possibility of leaching of metal ions from this adsorbent into the environment. However, the low desorption efficiency also suggests that the regeneration

capacity of the adsorbent is weak. Therefore, the unmodified SBC should not be used directly for water treatment, given the fine grain size. This raw material should be modified to make proper grain size granular and increase the adsorption capacity for As and heavy metals.

## 4. Conclusions

Iron-ore sludge (SBC) shows potential in the adsorption of As and heavy metals from water. The adsorption performance can be described by both the pseudo-first-order and the pseudo-second-order kinetic models, with a slightly better fit of the data to the latter, indicating that chemical adsorption occurred. The Langmuir maximum adsorption capacities of Pb, As, Cd, Zn, and Mn in the single-metal experiments were 1.305, 1.113, 0.771, 0.745, and 0.710 mg/g, respectively, and in the mixed-metal experiments for Pb, Cd, Zn, and Mn were 1.059, 0.484, 0.447 and 0.370 mg/g, respectively. The results show that iron-ore sludge is a promising adsorbent to remove potentially toxic pollutants from water.

**Author Contributions:** K.M.N. designed the research and wrote original draft preparation, B.Q.N. collected samples, conducted experiments, and analyzed samples, H.T.N. collected samples and conducted experiments; H.T.H.N. conceptualized, reviewed, and edited the paper.

**Funding:** This research is funded by Vietnam National Foundation for Science and Technology Development (NAFOSTED) under grant number 105.08-2017.02.

**Conflicts of Interest:** The authors declare no conflict of interest. The funders had no role in the design of the study; in the collection, analyses, or interpretation of data; in the writing of the manuscript, or in the decision to publish the results.

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
