# Peer review of "Adsorption of Arsenic and Heavy Metals from Solutions by Unmodified Iron-Ore Sludge"

_applsci, doi:10.3390/app9040619_

Round 1
Reviewer 1 Report
Overall Impressions:
The manuscript seems interesting to me. It is thoughtful and well-structured. However, I have some suggestions that, I think, would improve it.
First of all, I suggest to clear the paragraph in which you explain that “The adsorption process of tested contaminants was conducted in batch experiments of both single (As, Mn, Zn, Cd, and Pb) and mixed metal solutions.”…. How were the mixed solutions made?
The mixed metal solutions excluded As due to the possible reaction between cations and anions in the same solutions…Can you explain why you don`t have As into account more broadly? I think that It would be very enlightening if some bibliography about this point were included, too. Even more, this is a very important information, since normally in real waste water (for example in abandoned mine or other industries) As is one of the most common element.
Do you have SEM images of the iron mud after treatment? They could be useful to support the explanation of the adsorption process
Have you taking into account the effect of dosage? Trials could be carried out to study the dependence of adsorption on different doses of adsorbent.
Finally, I would suggest that, if the author could in the future, do some tests with real wastewater, this would prove the actual application of this waste.
Author Response
From: Dr. Ha T.H. Nguyen
Vietnam National University, Hanoi
334 Nguyen Trai, Thanh Xuan, Hanoi, Vietnam
Email: hoangha.nt@vnu.edu.vn
To: Reviewer #1 of the manuscript entitled “Adsorption of arsenic and heavy metals from solutions by unmodified iron ore sludge”
Thank you very much for your comments and suggestions. We revised our manuscripts in accordance with your comments, in which revised parts are expressed in red letters. Please see the following parts for our responses.
1. First of all, I suggest to clear the paragraph in which you explain that “The adsorption process of tested contaminants was conducted in batch experiments of both single (As, Mn, Zn, Cd, and Pb) and mixed metal solutions.”…. How were the mixed solutions made?
Response: Thank you so much for your suggestion. We forgot to describe this very basic and important information of our experiments. We added the information in lines 80-84.
2. The mixed metal solutions excluded As due to the possible reaction between cations and anions in the same solutions…Can you explain why you don`t have As into account more broadly? I think that It would be very enlightening if some bibliography about this point were included, too. Even more, this is a very important information, since normally in real waste water (for example in abandoned mine or other industries) As is one of the most common element.
Response: We totally agree with you that As is of great concern in water contamination in general and mining in particular. Our previous studies showed very high As concentrations in mining areas (both active and abandoned mines) with sulfur minerals formed from hydrothermal processes. Singe-metal experiments are very helpful to understand the mechanism; but for practical application, mixed-metal experiments should be performed. For these reasons, we did design the mixed-metal experiments including As at the first time. However, during the process of solution preparation for mixed-metal experiments, very tiny particles were observed (possibly due to precipitation). We analyzed the concentrations of As and heavy metals in adsorption kinetics; however, we could not get the desired initial metal concentrations (it is approximately 45-67% difference in comparison with the desired concentrations). Therefore, we decided to conduct other experiments with no As addition in the mixed-metal solutions. When writing this manuscript, we considered to remove As content in single-metal experiments; however, even we got failure with mixed-metal solutions with As, single-metal experiments with As may also provide some helpful information for adsorption capacity of this materials. It is not published yet, however, we already modified this material to make larger grain size and did the pilot experiment (5 m3/day) during 4 months using the direct wastewater from the largest Pb-Zn mine in Vietnam (As = 0.2-0.6 mg/l) and we got the expected result (it is not published).
Anyway, this is a limitation of our research. For mixed-metal experiments with As, it would be better if we used buffer solutions instead of simple dilution of nitrate salt solutions with Milli-Q water.
3. Do you have SEM images of the iron mud after treatment? They could be useful to support the explanation of the adsorption process
Response: We had SEM images of the material after treatment. However, because the material is sludge/mud with small grain size. Therefore, we could not take the same target particle before and after the experiment (only a mixture of particles). It is hard to observe the differences in SEM images before and after the experiments. Therefore, we did not insert the SEM images after treatment in our submitted manuscript.
4. Have you taking into account the effect of dosage? Trials could be carried out to study the dependence of adsorption on different doses of adsorbent.
Response: We also did this experiment because we needed that for design of other fixed-bed column experiments. We added the information in lines 86-87 and lines 177-186.
5. Finally, I would suggest that, if the author could in the future, do some tests with real wastewater, this would prove the actual application of this waste.
Response: As mentioned above, we also applied this material (modified) using the real wastewater at the largest Pb-Zn mine in Vietnam (Cho Don mine, Bac Kan Province). We also tried to bring wastewater from Cho Don mine (processing factory) to our laboratory; however, the concentrations of As and heavy metals decreased dramatically with time (pH of this wastewater ≈ 7) after 2-3 days, possibly the adsorption of metals onto suspended solids in the real wastewater (high contents of pyrite- FeS, chalcopyrite – FeAsS and other sulfur minerals in this mine).
Our revised manuscript was also edited by English editing service (https://www.stallardediting.com/; Code 18613). In the revised manuscript, English edited parts are expressed in blue letters.

Reviewer 2 Report
Some aspects to be corrected or revised in the manuscript:
1- In page 3, last line: the peak of 1031 cm-1 (please, use superscript when necessary) is shown in Figure 2 as 1029.79.
2- Authors explain the effect of pH as:"This can be explained by forming of hydroxide precipitates due to the increasing of OH- (superscript, please) concentration when the pH ≥ 7 was obtained,...". But in page 5, lines 146-157, there is a better explanation for this phenomenon, the influence of the pzc. And in page 5, lines 157-161, other good explanation: the competititon with protons. Really, both are descriptions of the same fenomenon. But, in my opinion, the forming of hydroxide precipitates is the weakest explanation, and, except for cadmium, I don't think there will be any precipitation even at pH=8.
Author Response
From: Dr. Ha T.H. Nguyen
Vietnam National University, Hanoi
334 Nguyen Trai, Thanh Xuan, Hanoi, Vietnam
Email: hoangha.nt@vnu.edu.vn
To: Reviewer #2 of the manuscript entitled “Adsorption of arsenic and heavy metals from solutions by unmodified iron ore sludge”
Thank you very much for your comments and suggestions. We revised our manuscripts in accordance with your comments, in which revised parts are expressed in red letters. Please see the following parts for our responses.
1. In page 3, last line: the peak of 1031 cm-1 (please, use superscript when necessary) is shown in Figure 2 as 1029.79.
Response: The FTIR parameters of the adsorbent have been revised in lines 139-140, including the superscript
2. Authors explain the effect of pH as: "This can be explained by forming of hydroxide precipitates due to the increasing of OH- (superscript, please) concentration when the pH ≥ 7 was obtained,...". But in page 5, lines 146-157, there is a better explanation for this phenomenon, the influence of the pzc. And in page 5, lines 157-161, other good explanation: the competition with protons. Really, both are descriptions of the same phenomenon. But, in my opinion, the forming of hydroxide precipitates is the weakest explanation, and, except for cadmium, I don't think there will be any precipitation even at pH=8.
Response: We agree with your suggestion. Several processes involved in adsorption capacity of As and heavy metals from the solutions. We revised and explained more detail this content in lines 168-173.
Our revised manuscript was also edited by English editing service (https://www.stallardediting.com/; Code 18613). In the revised manuscript, English edited parts are expressed in blue letters.

Reviewer 3 Report
This manuscript studies the adsorption of different metals ( As, Mn, Zn, Cd, and Pb) using iron ore steel slag. The subject is inside the scope of applied sciences since the authors investigate a physical process.
The novelty of the manuscript is low to moderate since there are hundreds of papers about the adsorption of different metals with different materials
Some points:
I cannot understand why the authors organize this way their manuscript. The main problem is the section Adsorption kinetics and then the section Adsorption Isotherms. Isotherms like Langmuir is kinetics. Usually, in this type of studies the researchers examine all the possible models, and then they choose the best fit, and they correlate the results with the mechanism
So form the manuscript not clear. The kinetics of the adsorption obeys Langmuir? Obeys the second-order kinetic model? The authors must compare all the available models and choose what they believe that represents the data and has a physical meaning
- What about the stability of the material? Have the authors tried to measure the leaching of the compounds that consist the material for different pH?
- What about the regeneration of the material? How is this possible in this type of materials?
Author Response
From: Dr. Ha T.H. Nguyen
Vietnam National University, Hanoi
334 Nguyen Trai, Thanh Xuan, Hanoi, Vietnam
Email: hoangha.nt@vnu.edu.vn
To: Reviewer #3 of the manuscript entitled “Adsorption of arsenic and heavy metals from solutions by unmodified iron ore sludge”
Thank you very much for your comments and suggestions. We revised our manuscripts in accordance with your comments, in which revised parts are expressed in red letters. Please see the following parts for our responses.
1. The novelty of the manuscript is low to moderate since there are hundreds of papers about the adsorption of different metals with different materials
Response: We agree with you that a variety of adsorbents have been reported. However, there are few studies on the capacity of sludge from iron-ore processing to adsorb As and heavy metals for water treatment. This is an environment-friendly and cost-effective technique that uses solid wastes for water treatment and approaching advantageous idea of waste-free production.
2. I cannot understand why the authors organize this way their manuscript. The main problem is the section Adsorption kinetics and then the section Adsorption Isotherms. Isotherms like Langmuir is kinetics. Usually, in this type of studies the researchers examine all the possible models, and then they choose the best fit, and they correlate the results with the mechanism. So form the manuscript not clear. The kinetics of the adsorption obeys Langmuir? Obeys the second-order kinetic model? The authors must compare all the available models and choose what they believe that represents the data and has a physical meaning.
Response:
- We agree with you that “adsorption kinetics” and “adsorption isotherns” are only concepts that are commonly used in many adsorption studies that express the different experimental designs and common models used. We also agree that several available models are used and suitable model(s) is/are selected to explain the mechanisms.
- Adsorption kinetics: study the effect of time (in our study several intervals from 10 minutes to 1440 minutes were designed) on metal adsorption. To the best of our knowledge, under review of papers, pseudo-first-order and pseudo-second-order are commonly used for adsorption kinetics. In our study, adsorption kinetic experiments were conducted to define the rate of metals adsorption onto the adsorbent, we found the better model was the pseudo-second-order and then indicated chemisorption as adsorption mechanism.
- Adsorption isotherms: study the effect of initial concentrations of metals in experimental solutions (in our study metal concentrations from 0 to 50 mg/L per each metal were designed). The isotherm experiments (or equilibrium adsorption) were performed after kinetic experiments, after having basically grasped the time that the adsorbent reached saturation state. In other words, kinetic are initial experiments, providing the basis for selecting a 24-hour experimental time in isotherm. We revised the title 3.5 in line 212 to “batch equilibrium adsorption” to indicate the different experiments for kinetics and equilibrium adsorption. To the best of our knowledge, under review of papers, Langmuir and Freundlich are commonly used for adsorption isotherm or equilibrium adsorption. The data of our study fit to the Langmuir model, and we have continued to use the parameters of this model for next logical explanations.
These two experiments are basic designs for adsorbent study that we refer from many previous studies (Vasanth Kumar and Sivanesan, 2006; Kuo et al., 2008; Lima et al., 2015; Tran et al., 2017). These two experiments were not only designed for mechanism but also providing very important input information (including time, initial concentrations of metal in inlet water, and dose of adsorption) for further design and applications of the adsorbent in fixed-bed column, pilot, and field scale. Therefore, we would like to describe two experiments separately in our revised manuscript.
- We also agree that the conclusion of the adsorption mechanism based on these models is just a predictive statement. Adsorption mechanisms have to be established by (1) using several analytical techniques (i.e., FTIR, SEM, nitrogen adsorption-desorption isotherms, Raman spectroscopy, TGA/DTA, DSC, 29Si and 13C solid-state NMR, XRD, XPS, pHPZC, solution calorimetry, ...) and (2) having a good sense of the adsorbate and adsorbent (Volesky, 2007; Lima et al., 2015, Oladoja, 2016).
Please see our mentioned references as follows:
- Vasanth Kumar, K., Sivanesan, S. (2006). Equilibrium data, isotherm parameters and process design for partial and complete isotherm of methylene blue onto activated carbon. J. Hazard. Mater. 134 (1-3), 237-244
- Kuo, C. Y., Wu, C. H., & Wu, J. Y. (2008). Adsorption of direct dyes from aqueous solutions by carbon nanotubes: Determination of equilibrium, kinetics and thermodynamics parameters. Journal of Colloid and Interface Science, 327(2), 308-315.
- Tran, H. N., Wang, Y. F., You, S. J., & Chao, H. P. (2017). Insights into the mechanism of cationic dye adsorption on activated charcoal: the importance of π–π interactions. Process Safety and Environmental Protection, 107, 168-180.
- Volesky, B., 2007. Biosorption and me. Water Res. 41 (18), 4017-4029
- Lima, E.C., Adebayo, M.A., Machado, F.M. (2015). Kinetic and Equilibrium Models of Adsorption. In: Carbon Nanomaterials as Adsorbents for Environmental and Biological Applications. Springer, pp. 33-69
- Oladoja, N. A. (2016). A critical review of the applicability of Avrami fractional kinetic equation in adsorption-based water treatment studies. Desalination and Water Treatment, 57(34), 15813-15825.
3. What about the stability of the material? Have the authors tried to measure the leaching of the compounds that consist the material for different pH?
What about the regeneration of the material? How is this possible in this type of materials?
Response:
- For practical use of this adsorbent, we also did the experiments on the leaching possibility of As and heavy metals from adsorbents at normal range of pH in mining areas in Vietnam (pH = 4.0–9.0). These experiments were conducted as soon as possible after the isotherm experiments (multi-metals). Our experimental result showed low desorption capacity. Regeneration of materials is considered very important criteria for re-use of adsorbents. In our study, adsorbent is from unmodified iron ore sludge and should not be directly used for water treatment. Our study focus on providing adsorption capacity of potential adsorbent from iron ore sludge as raw material for further modification (e.g., making proper granular, increasing adsorption capacity) and application.
- We added this information in lines 108-111 (methods) and lines 254-270 (results and discussion)
Our revised manuscript was also edited by English editing service (https://www.stallardediting.com/; Code 18613). In the revised manuscript, English edited parts are expressed in blue letters.

Round 2
Reviewer 3 Report
The authors revised their manuscript according to the suggestions